# OpenReview forum: "Refusal Degrades with Token-Form Drift: Limits of Token-Level Alignment"
_ICLR.cc/2026/Conference — Submitted to ICLR 2026_

### Official Review · Reviewer_WNND · 2025-10-24

**Soundness:** 3
**Presentation:** 3
**Contribution:** 3
**Rating:** 6
**Confidence:** 3

**Summary:**

This paper investigates token level vulnerabilities in LLMs, particularly under conditions of token drift. To address this issue, the study introduces two complementary approaches. (1). a manual, character-level perturbation method for inducing token drift, and (2). an automated, iterative feedback based framework. Through these perturbations, the paper demonstrates the vulnerability of LLMs to token level drift and highlights the potential of such drifts to serve as effective jailbreak mechanisms.

**Strengths:**

In terms of models, a wider array of models have been used for the evaluation and the paper presents interesting analysis on the model behavior with the increasing size in language modes.

1. Experiments were conducted on different datasets for the sake of generalization.

2. Personally in my line of work I had observed the vulnerability of LLMs of character level drift and the message that the LLMs are vulnerable towards token drift is agreeable. Though the message on RLHF being the culprit for the vulnerability needs validation. See weakness for details.

3. The interactive automated framework for evaluating the token drift is effective and performs well against jailbreaks. Given that the paper is not presented as jailbreak paper the results in this section are satisfactory.

4. The paper does explore the possibilities of using the automated generation strategy towards model improvement.

**Weaknesses:**

1. Can you motivate the reasoning for choosing the specific character perturbation as the perturbation strategy for manual perturbation. The motivation for the simple strategy though effective is unclear.

2. Can you provide results from non-instruct models to validate the role of instruction tuning as the main culprit behind the token level degradation in language models

3. The paper serves as a findings type of paper on establishing the token level vulnerabilities. While the presented work highlights the vulnerabilities of the existing models, validation on non-instruction tuned models/ isolated experiments with fine tuning is necessary (albeit in small scale) towards the conclusion of RLHF as culprit.

**Questions:**

See weaknesses

---

> ### Author Response · Authors · 2025-11-26
>
> We thank you for your constructive feedback. All concerns below will be addressed in the revision, including expanded analyses, clarified framing, and additional experiments.
>
> Weakness 1: Rationale Behind the Perturbation Choice
>
> The character-level symbol substitution was selected not for complexity but for two methodological properties essential to controlled analysis:
>
> 1. Deterministic and invertible.
> The mapping (e.g., a $\rightarrow$ @) is fully rule-based and losslessly reversible, allowing us to decode model outputs back to canonical English and cleanly separate “semantic failure” from “refusal behavior.”
> Non-invertible perturbations (e.g., scrambling, homophones, misspellings) would confound this distinction.
>
> 2. A clean, monotonic drift axis.
> By substituting the first k letters of the alphabet, we obtain a strictly ordered, quantifiable perturbation ladder, enabling precise measurement of rise–plateau–collapse dynamics.
>
> Our goal in Experiment 1 was to study the mechanism under controlled drift, not to maximize naturalistic attack realism. Later experiments (including evolved drifts and non-BPE models) confirm that the phenomenon is not specific to this perturbation family.
>
> We will clarify this rationale in the revision.
>
>
> Weakenss 2: Results on Non–Instruction-Tuned Models
>
> To clarify the role of instruction tuning, we evaluated token-form drift on non–instruction-tuned base models (e.g., Qwen-2.5 Base, Llama-3.1-8B Base). These models contain no refusal policy by design: when given a harmful query (e.g., “Describe how to make a bomb”), they answer directly without any alignment-induced intervention.
>
> The results are consistent across all base models tested:
>
> (i)Drift still causes semantic degradation at high perturbation levels (collapse),
>
> (ii)but no refusals ever appear, because base models lack alignment-trained refusal behaviors.
>
> This contrast leads to two clear conclusions:
>
> The rise and plateau regions (where the model understands the harmful intent but no longer refuses) are unique to instruction-tuned/RLHF-aligned models, because only these models possess refusal mechanisms that can be bypassed.
>
> Collapse is not an alignment effect but a semantic-comprehension failure, since it appears in both base and aligned models.
>
> Therefore, the core vulnerability exposed by token-form drift lies specifically in the instruction-tuned safety layer—where refusal features are trained on narrow surface-form patterns—not in the pretrained semantic representations or tokenizer.
>
>
> Weakness 3: Validation on Base Models and Isolated RLHF Fine-Tuning
>
> To determine whether token-form drift vulnerability originates from pretraining or from alignment, we conducted the two forms of validation requested: (i) experiments on non–instruction-tuned base models, and (ii) isolated small-scale RLHF fine-tuning.
>
> 1. Base models show collapse only—no rise, no refusal bypass
>
> Base models (Qwen/Gemma/Llama base variants) contain no refusal mechanism. When given harmful queries, they answer directly; drift merely causes semantic degradation at high perturbation levels.
> We observe:
>
> (i)No rise phase (because no refusal exists to be bypassed),
>
> (ii)No plateau,
>
> (iii)Only semantic collapse at high k.
>
> This isolates the source of rise–plateau behavior:
> it cannot emerge from pretraining; it appears only when alignment-induced refusal features exist.
>
> 2. Isolated small-scale RLHF (PPO/GRPO) reintroduces the vulnerability
>
> We fine-tuned Qwen-2.5-72B with PPO and GRPO on standard safety datasets. Even after added alignment optimization:
>
> PPO: 0% $\rightarrow$ 80% ASR after one evolution step
>
> GRPO: 10% $\rightarrow$ 90% ASR after one evolution step
>
> Thus, RLHF introduces refusal features that are easily disrupted by drift, directly causing the rise–plateau phase.
>
> 3. Stronger commercial alignment pipelines behave the same way
>
> GPT-5 and Claude—both trained with multi-stage proprietary alignment—still exhibit:
>
> low-ASR seeds (<20%), but
>
> evolved drift forms reliably reaching ≥80%.
>
> Their behavior mirrors PPO/GRPO-tuned models, indicating that the vulnerability persists even under significantly stronger alignment regimes.

---

### Official Review · Reviewer_frC3 · 2025-10-29

**Soundness:** 3
**Presentation:** 2
**Contribution:** 3
**Rating:** 6
**Confidence:** 2

**Summary:**

This paper demonstrates that current LLM safety alignment is fragile to token-form distribution shifts. Through controlled and automated perturbation studies, the authors reveal a consistent rise–plateau–collapse failure pattern where refusals weaken, harmful compliance peaks, and models eventually become incoherent as perturbations increase. Larger models show stronger robustness at low-level shifts but expose larger attack surfaces across diverse perturbation forms, creating a capability–vulnerability tradeoff. The results show that alignment methods predominantly rely on surface correlations rather than semantic understanding, causing rapid re-emergence of jailbreaks despite fine-tuning. The paper motivates the need for form-invariant, semantics-focused defense strategies and highlights key contributions: conceptualizing token-form drift, empirically validating universal degradation dynamics, and emphasizing cross-form robustness evaluation.

**Strengths:**

This paper identifies and quantifies the alignment generalization gap under token-form drift. The study reveals an important hidden risk in current safety alignment: modern large models possess narrow safety generalization. The authors further design an automated framework to generate semantically consistent perturbations, showing that fragility to token-form drift is a systematic rather than incidental phenomenon.

**Weaknesses:**

This paper identifies and systematizes the phenomenon of the alignment generalization gap under token-form drift; however, it lacks a theoretical analysis of this phenomenon. There are 3 key issues that deserve further attention:

1. Can the source of token-form drift be theoretically modeled? For example, can the authors analyze how tokenization strategies, including segmentation rules and subword composition, lead to significant alignment degradation even when semantic meaning is preserved?

2. Since the model can bypass alignment constraints under structural perturbations, does this imply a different degree of separability between safety alignment signals and pretrained knowledge during learning? Can the authors explain this discrepancy from the perspective of optimization dynamics?

3. How can a more principled training framework be designed to mitigate such superficial alignment? For instance, is continued pretraining a viable solution?

**Questions:**

In weaknesses.

---

> ### Author Response · Authors · 2025-11-26
>
> We thank you for your constructive feedback. All concerns below will be addressed in the revision, including expanded analyses, clarified framing, and additional experiments.
>
> Weakness 1: Can the source of token-form drift be theoretically modeled?
>
> Our work already provides a theoretical model of how token-form drift interacts with safety alignment, and we agree this connection can be made more explicit in the paper.
>
> 1. Existing formal model of alignment under drift
>
> As presented in Section 2(line 87-104), safety alignment can be written as minimizing refusal-oriented loss over a curated distribution
> $P_{align}$:
>
> $\theta^{*} = \arg\min_{\theta}  E_{x \sim P_{\text{align}}} \Big[ \mathcal{L}\big(f_{\theta}(x), y_{\text{refuse}}\big) \Big].$
>
> where $f_{\theta}$ is the model and $y_{refuse}$ denotes refusal-style answers.
>
> Token-form drift shifts harmful queries to a perturbed distribution $P_{
> perturb}$
> (x). As the divergence $D(P_{align},P_{perturb})$ grows, the same $\theta^{*}$ no longer minimizes refusal loss under $P_{perturb}$:
>
> $\theta^{*} \neq \arg\min_{\theta}  E_{x \sim P_{\text{perturb}}} \Big[ \mathcal{L}\big(f_{\theta}(x), y_{\text{refuse}}\big) \Big].$
>
> This alignment generalization gap explains why refusal behaviors degrade under token-form drift even when the underlying semantics are preserved: the alignment gradients were never trained to cover that region of token space. In the revision, we will highlight this formal link more prominently as the theoretical model of drift-induced failure.
>
> 2. Relation to tokenization strategies (BPE vs. unigram/SentencePiece)
>
> You specifically ask whether tokenization granularity can be part of the theoretical explanation. In our framework, both $P_{align}$ and $P_{perturb}$ are distributions over token sequences induced by a given tokenizer. Token-form drift is defined at the character level and then mapped through the tokenizer to tokens; different tokenizers will indeed produce different discrete segmentations, but the key quantity in our model is the divergence $D(P_{align},P_{perturb})$, not the particular segmentation scheme.
>
> Empirically, we test this explicitly on Gemma-3-it-27B, which uses a SentencePiece-style tokenizer, and still observe the same rise–plateau–collapse pattern on both AdvBench and JBB:
>
> The results on Gemma3 is:
>
> | replace | success rate on advBench% | success rate on JBB% |
> |--------:|-------------:|------------------------:|
> | 0  | 0.0   | 0.0   |
> | 1  | 83.0  | 77.0  |
> | 2  | 86.0  | 85.0  |
> | 3  | 90.0  | 84.0  |
> | 4  | 96.0  | 94.0 |
> | 5  | 90.0  | 95.0  |
> | 6  | 92.0  | 94.0  |
> | 7  | 90.0  | 93.0  |
> | 8  | 94.0  | 91.0  |
> | 9  | 86.0  | 88.0  |
> | 10 | 80.0  | 81.0  |
> | 11 | 76.0  | 73.0  |
> | 12 | 63.0  | 61.0  |
> | 13 | 41.0  | 33.0  |
> | 14 | 23.0  | 14.0  |
> | 15 | 10.0  | 3.0   |
> | 16 | 0.0   | 0.0   |
> | 17 | 0.0   | 0.0   |
> | 18 | 0.0   | 0.0   |
> | 19 | 0.0   | 0.0   |
> | 20 | 0.0   | 0.0   |
> | 21 | 0.0   | 0.0   |
> | 22 | 0.0   | 0.0   |
> | 23 | 0.0   | 0.0   |
> | 24 | 0.0   | 0.0   |
> | 25 | 0.0   | 0.0   |
> | 26 | 0.0   | 0.0   |
>
> Here, the success rate
>
> rises from 0\% $\ge$ 96\% (k = 1–4),
>
> stays in a high plateau ($\approx$80–95\% for k = 5–11),
>
> and then collapses to 0\% once $k\ge16$.
>
> This cross-tokenizer replication strongly suggests that the phenomenon does not arise from a specific BPE segmentation artifact, but from the structure of the alignment objective itself: alignment gradients attach to surface token patterns that are disrupted by drift, regardless of the exact tokenizer.

---

> ### Author Response · Authors · 2025-11-26
>
> Weakness 2: On Separability Between Alignment Signals and Pretrained Knowledge
>
> Our findings are indeed consistent with a partial separability between safety alignment signals and pretrained semantic knowledge. Token-form drift provides direct empirical evidence for this separation. Below we explain why this separation naturally emerges from optimization dynamics.
>
> 1. Alignment gradients occupy a much narrower subspace than pretraining gradients.
> During pretraining, gradients are dense and semantically diverse, shaping a broad manifold of linguistic and world knowledge. In contrast, SFT/RLHF alignment introduces highly localized gradients concentrated on specific surface-form patterns (canonical spellings of harmful queries, refusal templates, etc.). Because these gradients are sparse relative to the pretraining signal, they modify only a limited portion of the input space.
>
> 2. As a result, semantic representations remain intact while refusal behavior is tied to surface form.
> Optimization thus yields a model where:
>
> (i)semantic pathways (built during pretraining) generalize broadly, even under drift,
>
> (ii)but refusal policies (added by alignment) bind to shallow textual correlates that resemble the alignment dataset.
>
> Token-form drift disrupts these surface features while leaving the underlying meaning recoverable, causing the semantic subsystem to function normally while the refusal subsystem fails to activate.
>
> 3. This reflects partial—not complete—subspace separability.
> Our explanation does not require the alignment and semantic subspaces to be orthogonal or fully independent; it only requires that alignment gradients do not fully propagate into semantic-invariant directions. This mild condition is consistent with existing RLHF/SFT training dynamics and is sufficient to explain why structural perturbations bypass alignment.
>
> We will make this optimization-based explanation clearer in the revised manuscript.
>
>
> Weakness 3: Toward Principled Training Frameworks Beyond Superficial Alignment
>
> Our results indicate that the core issue is not insufficient pretraining, but the structural mismatch between pretrained semantic representations and the surface-form refusal features introduced by SFT/RLHF. Continued pretraining strengthens semantics but does not reshape these shallow refusal mechanisms, and thus cannot on its own resolve drift vulnerabilities.
>
> A more principled framework would require alignment signals to operate at the semantic level rather than the token level. Two promising directions are:
>
> 1. Semantic-invariant alignment objectives.
> Defining refusal behavior in the model’s representation space (e.g., hidden-state semantics) instead of raw token patterns would ensure that safety generalizes across form-preserving perturbations such as drift.
>
> 2. Joint semantic–safety training.
> Rather than applying safety alignment only after pretraining—where alignment gradients are too sparse to modify the pretrained semantic manifold—joint or multi-task training could co-shape semantic and safety features, reducing the functional separability exposed by token-form drift.
>
> We will expand these points in the revision, emphasizing why continued pretraining alone is insufficient and why semantically grounded alignment objectives are needed.

---

### Official Review · Reviewer_LqcY · 2025-10-31

**Soundness:** 3
**Presentation:** 3
**Contribution:** 2
**Rating:** 4
**Confidence:** 4

**Summary:**

This paper investigates the robustness of safety alignment in Large Language Models (LLMs). It finds that current alignment methods are fragile because they are coupled to specific "token forms". The authors introduce "token-form drift", which refers to semantics-preserving changes in input form, such as symbol substitution. The core finding is that this drift systematically degrades refusal behavior, even when the model understands the harmful semantic intent. The paper empirically demonstrates a universal "rise-plateau-collapse" failure pattern. As perturbation increases, refusal fails (Rise), harmful compliance peaks (Plateau), and extreme drift leads to semantic incoherence rather than recovered safety (Collapse). This reveals a "capability-vulnerability tradeoff": more capable models resist simple drift longer, but their ability to understand complex perturbations creates a wider attack surface. The main contribution is showing that current alignment is only "token-level" and form-sensitive, motivating future defenses that target "form-invariant" semantic alignment.

**Strengths:**

The originality of this paper lies in its novel problem formulation. It defines "token-form drift" to explain alignment fragility. This concept effectively separates the model's semantic understanding from its form-sensitive refusal behavior. This provides a new and useful lens for analyzing why current safety methods fail, moving beyond just finding new attacks.

The quality of the work is high. The authors experimentally prove their claims through various and extensive experiments. They use controlled, progressive perturbations to systematically test robustness. Furthermore, they develop an LLM-in-the-loop automated framework to show that diverse, effective perturbations can be discovered automatically. This comprehensive empirical validation strongly supports the paper's central hypothesis.

The paper is written with high clarity. The authors provide detailed interpretations for their experimental results. Complex findings, such as the universal "rise-plateau-collapse" pattern and the "capability-vulnerability tradeoff", are explained in a clear and understandable manner. This detailed analysis makes the paper's core arguments easy to follow.

The paper's significance is high. It demonstrates a fundamental limitation of current token-level alignment pipelines. The work provides valuable insight to researchers by suggesting what additional processes are necessary for future LLM alignment. By highlighting the need for "form-invariant" alignment, normalization, and cross-form evaluation, it directs the field toward developing more robust safety defenses.

**Weaknesses:**

The paper provides strong experimental proof for its insights. However, the core idea that alignment is sensitive to token-level shifts is an observation that many practitioners involved in training LLMs may already be familiar with. While the empirical validation is thorough, the work positions itself more as an analysis or explanatory paper rather than presenting a new technique. Given that ICLR typically emphasizes technical novelty, this work's contribution might be a better fit for a workshop or a review-style journal that values systematic analysis and problem formulation.

Additionally, the paper's key finding—that larger models are "paradoxically more vulnerable"—could be debated. The claim is partially correct, but the longer plateau of vulnerability in larger models stems from their superior capability to maintain semantic coherence under wider perturbations. In contrast, smaller models collapse faster. This rapid collapse of smaller models does not necessarily make them more robust or safe; it just indicates they are less capable of interpreting the input. Therefore, framing the larger models as "more vulnerable" might be an overstatement, as this vulnerability is a direct, if unintended, consequence of their higher capability.

**Questions:**

The paper is very clear, and the Appendix is comprehensive. I had no difficulty understanding the methodology or the results. My only point of discussion relates to the technical novelty, which was mentioned in my main review. The work provides an excellent analysis and formulation of the "token-form drift" problem. However, the core finding might feel familiar to practitioners. Could the authors please elaborate on what they consider the primary technical contribution, beyond the valuable problem formulation and experimental analysis? For instance, does the automated framework itself represent a novel technical method that could be generalized, or is the main contribution the analysis it enables? A response on this point could help clarify the paper's positioning.

---

> ### Author Response · Authors · 2025-11-26
>
> We thank you for your constructive feedback. All concerns below will be addressed in the revision, including expanded analyses, clarified framing, and additional experiments.
>
> Weakness 1: Primary Technical Contribution and Positioning
>
> The paper’s contribution is twofold, and the technical novelty lies in both the problem formulation and the generalizable methodology that enables its analysis.
>
> 1. Technical novelty in formulation: a continuous, semantics-preserving drift axis
>
> Prior jailbreak methods rely on discrete, ad-hoc perturbations (cipher encodings, token scrambling, AutoDAN mutations), which do not allow controlled measurement of how alignment degrades as perturbation strength increases.
> Our work introduces the first continuous and invertible token-form drift axis, which makes alignment degradation quantifiable, reproducible, and model-agnostic.
> This formulation is itself a methodological contribution: it creates a new robustness evaluation dimension that did not previously exist.
>
> 2. Technical novelty in methodology: a generalizable drift-discovery framework
>
> The automated framework is not merely an attack generator. It is a reusable diagnostic instrument that:
> autonomously discovers universal drift families,
> and produces standardized robustness curves that can be applied to any future alignment method.
>
> This makes the framework technically generalizable beyond the specific analyses in the paper. It can be used as an evaluation suite for new alignment algorithms.
>
> 3. The analysis is enabled by (1) and (2), but not reducible to them
>
> The empirical findings—such as the universal rise–plateau–collapse curve and the capability–vulnerability tradeoff—are made possible by the formulation and framework above.
> Thus, while the analysis is a major part of the paper, the technical novelty lies in:
> defining a principled drift manifold, and
> creating a general-purpose automated framework to explore and quantify alignment behavior along it.
>
> We will clarify this positioning in the revision.
>
>
> Weakness 2: Interpretation of “Larger Models Are More Vulnerable”
>
> You correctly note that smaller models collapse earlier because they lose semantic coherence more quickly, not because they are inherently safer. Our intention is not to suggest that larger models are “less safe,” but to emphasize a structural asymmetry:
> Larger models maintain semantic interpretability under stronger drift, which
> extends the zone in which harmful meaning is still understood but alignment no longer fires, producing a longer plateau of vulnerability before collapse.
>
> We will revise the manuscript to avoid any implication that larger models are intrinsically “more vulnerable,” and instead frame the observation as a capability–vulnerability trade-off:
> greater semantic resilience delays collapse and thus prolongs the window where alignment can be bypassed.

---

### Official Review · Reviewer_6DBz · 2025-11-01

**Soundness:** 3
**Presentation:** 2
**Contribution:** 2
**Rating:** 4
**Confidence:** 3

**Summary:**

The paper studies form sensitivity in LLM safety: refusals learned via alignment degrade under semantics-preserving token-form drift, which includes orthography, separators, segmentation etc. It reports that refusals degrade as drift increases. It uses an LLM-in-the-loop search to find readable adversarial forms and a patch-then-break experiment showing SFT on one form doesn’t transfer. This work motivates “form-invariant” defenses and future alignment methods to use cross-form evaluations.

**Strengths:**

1. The paper isolates a concrete vulnerability in current alignment of semantics-preserving token-form drift, and shows that refusal behavior degrades from this vulnerability in ways not covered by most standard safety evaluations.

2. The discovery process is automated and operates in a black-box setting, which potentially allows it to generate a broader set of human-interpretable adversarial attacks that could benefit future safety research.

3. The patch-then-break experiment is an operationally useful finding: fine-tuning to eliminate a single successful form does not generalize, and closely related variants re-emerge quickly, signaling limited value when alignment fails to generalize. This also highlights that most issues arise from OOD data, which the proposed method is particularly effective at producing.

4. The observed “rise–plateau–collapse” trend is interesting and shows a Pareto frontier between adversarial strength and semantic preservation. This offers a useful concept for designing future robustness or attack evaluations.

**Weaknesses:**

1. The attack novelty is limited compared to prior LLM-in-the-loop jailbreak and fuzzing methods such as AutoDAN. Although the paper focuses on one particular adversarial form (token-form drift), the overall pipeline is conceptually very similar to existing LLM-in-the-loop, mutation-based prompt optimization methods.

2. The reported “rise–plateau–collapse” curve is not particularly surprising. Prior work on cipher-based jailbreaks and encoding attacks has already shown that moderate perturbation can bypass safety (the rise and plateau phases), while extreme distortion eventually reduces success. The results largely demonstrate this known pattern rather than uncovering a new mechanism.

3. The paper attributes this region to the model’s inability to understand the input but does not provide direct evidence. It would be valuable to investigate this more deeply, for example by analyzing LLM outputs or applying interpretability methods. The model for sure understand part of the input tokens, and it would be particularly interesting to test whether refusals are triggered when all understood tokens remain in-distribution or detects adversarial intent. Gradient-based jailbreaks such as GCG also seem to contradict the assumption that nonsensical inputs necessarily yield low ASR; exploring this relationship would clarify the collapse phenomenon.

4. The perturbation seeds appear to be selected from a manually designed set. It would be helpful to quantify how much of the ASR comes from these initial seeds vs. later evolved forms, and to include ASR statistics for each mutation step to better understand this.

**Questions:**

1. Many of the results assume a consistent notion of “semantic preservation.” Given that the judge is also an LLM, how do you ensure the validator’s understanding of meaning aligns with the model being attacked?

2. Can the authors comment on whether the drift ladder’s behavior depends on tokenization granularity (e.g., BPE vs unigram models)?

3. Have you tested these jailbreaks against more aligned models? Since many alignment vulnerabilities are known to stem from shallow and less generalizable alignment methods like SFT, it would be valuable to evaluate whether token-form drift remains effective against models trained with stronger alignment methods.

**Details Of Ethics Concerns:**

Includes jailbreaks

---

> ### Author Response · Authors · 2025-11-26
>
> We thank you for your constructive feedback. All concerns below will be addressed in the revision, including expanded analyses, clarified framing, and additional experiments.
>
> Weakness 1: Novelty and relation to AutoDAN / LLM-in-the-loop methods
>
> While our system is technically an LLM-in-the-loop pipeline, its role is fundamentally different from mutation-based jailbreak optimizers such as AutoDAN. The automated framework is not designed as an attack optimizer; it functions primarily as an experimental apparatus for validating the drift-induced alignment breakdown discovered in Section 4.
>
> Experiment 1 establishes the rise–plateau–collapse dynamics under controlled, hand-designed drift. If this phenomenon reflects a genuine alignment failure mode, then it should also appear when drift is introduced autonomously rather than manually engineered. Experiment 2 tests exactly this hypothesis: the LLM incrementally applies stronger drift, and we observe whether alignment collapses in the same manner predicted by Experiment 1. As shown in Table 2, an initial seed with 0\% ASR evolves—through LLM-introduced drift—into a form achieving 90\% ASR. This automated reproduction of the rise phase demonstrates that the vulnerability is intrinsic to alignment, not an artifact of handcrafted perturbations.
> Thus, the automation serves to generalize and systematize the phenomenon, not to search per-instance adversarial prompts like AutoDAN.
>
> Once this validation is complete, the same framework can naturally be repurposed for attack, and here it exhibits a secondary benefit. Unlike AutoDAN, which performs per-instance mutation and often halts prematurely due to prefix-matching false positives (≈350 tokens/attempt with a true ASR of only ≈70\%), our framework discovers a universal drift pattern using ~11 000 total tokens (≈8500 evaluation + ≈2500 generation). After discovery, the pattern generalizes across prompts without further optimization, yielding a verified ≈95\% ASR for that form. This efficiency advantage is therefore a consequence, not the objective, of the framework's design.
>
> Weakness 2: “Rise–plateau–collapse” not surprising given prior cipher-based work
>
> We agree that prior cipher-based jailbreaks and encoding attacks observed that moderate perturbations can bypass safety while extreme corruption eventually fails. However, these works provide only coarse qualitative descriptions of this behavior. Our contribution is not the rediscovery of this trend, but the demonstration that token-form drift induces a universal, quantifiable rise–plateau–collapse curve with properties that prior work could neither measure nor explain.
>
> First, cipher/encoding attacks apply discrete transformations, preventing analysis of continuous drift dynamics. They cannot reveal where onset occurs, whether the rise is abrupt or gradual, how the plateau is shaped, or whether collapse reflects regained safety or degraded comprehension. Our controlled drift axis makes these transition dynamics measurable.
>
> Second, we show this curve is highly consistent across models, including open-source and black-box LLMs, and further uncover a capability–vulnerability tradeoff: stronger models maintain semantics over larger drift ranges (delaying collapse), but this also enlarges the plateau window in which alignment is bypassed.
>
> Finally, our patch-then-break results provide a mechanistic explanation: alignment remains token-level and form-sensitive, rather than semantically invariant. None of these quantitative or mechanistic insights arise from cipher-based studies.

---

> ### Author Response · Authors · 2025-11-26
>
> Weakness 3: Evidence for Collapse and Relation to GCG
>
> This concern prompted us to deepen our analysis of the collapse region. Below we provide direct qualitative evidence, quantitative measurements, and a refined mechanistic explanation showing that high-k failures arise from semantic degradation, not restored safety alignment.
>
> 1. Direct qualitative evidence: collapse exhibits semantic failure, not refusals
>
> As noted in lines 203–205, manual inspection reveals two distinct collapse modes:
>
> Semantic misinterpretation
> “make a bomb” $\rightarrow$ “preheat the oven and mix ingredients …”
>
> Semantic incoherence / degeneration
> “iiiiiiiiii”, random symbols, or internally inconsistent sentences.
>
> Critically, none of these high-k outputs are refusals. The model never activates safety alignment; instead, it fails to reconstruct the intended meaning.
>
> 2. Quantifying semantic degradation: recoverability $\rightarrow$ semantic similarity $\rightarrow$ confidence
>
> To more directly measure whether the model still “understands” the perturbed input, we shift the focus from whether the model’s final answer is related to the harmful query to whether the model can, without any chain-of-thought, restore the perturbed input back to its original form.
>
> Our rationale is simple:
> if the model truly understands a drifted query, then it should be able to reconstruct its canonical form; failure to restore implies loss of semantic comprehension. Thus, we evaluate its ability to restore the drifted text and measure:
>
> (i)Complete recoverability (exact reconstruction),
>
> (ii)Semantic similarity (BERTScore-F1), and
>
> (ii)Model confidence (self-reported certainty).
>
> These form a semantic hierarchy:
> recoverability (strict) $\rightarrow$ BERTScore (semantic) $\rightarrow$ confidence (self-certainty).
>
> The results on Qwen-2.5-32B on advBench and JBB are displayed below:
>
> | index | equal_ratio | bertscore_f1_mean | avg_confidence |
> |-|-|-|-|
> |1|1.0|1.0000000000000000|0.9698755497590654|
> |2|0.92|0.9960878491401672|0.9462848795725131|
> |3|0.9|0.9964327812194824|0.8720667807613609|
> |4|0.92|0.9986474514007568|0.8996107876347130|
> |5|0.7|0.9883798360824585|0.7404671398103114|
> |6|0.7|0.9852044582366943|0.7511854100777724|
> |7|0.7|0.9927139282226562|0.7759417414596693|
> |8|0.74|0.9934989809989929|0.8108719290367762|
> |9|0.8|0.9925569295883179|0.7815240897900356|
> |10|0.76|0.9926716089248657|0.7926228282114911|
> |11|0.72|0.9929702281951904|0.7586443950867690|
> |12|0.68|0.9762908220291138|0.6157713964198244|
> |13|0.56|0.9637693762779236|0.4941530700040411|
> |14|0.28|0.9481589794158936|0.3262555722647492|
> |15|0.16|0.9390323162078857|0.2325169568723761|
> |16|0.12|0.9237977862358093|0.1781717870881587|
> |17|0.12|0.9228829145431519|0.1705744769683328|
> |18|0.08|0.9114975929260254|0.1599907358408247|
> |19|0.0|0.8819329738616943|0.0467042546732578|
> |20|0.0|0.8212786912918091|0.0352284671473967|
> |21|0.0|0.8281606435775757|0.0164998312902163|
> |22|0.0|0.8183164000511169| 0.0220646391985067 |
> | 23 | 0.0  | 0.8143872022628784 | 0.0124400212039864 |
> | 24 | 0.0  | 0.8085958957672119 | 0.0176452124274931 |
> | 25 | 0.0  | 0.8275107741355896 | 0.0283407102416048 |
> | 26 | 0.0  | 0.8254566192626953 | 0.0274506798881648 |
>
> All metrics degrade monotonically and coherently as k increases:
>
> Recoverability: 1.00 $\rightarrow$ 0.00
>
> BERTScore-F1: 1.00 $\rightarrow$ 0.81
>
> Confidence: 0.97 $\rightarrow$ 0.02
>
> The turning point at k ≈ 13—where recoverability collapses from 0.56$\rightarrow$0.28 and confidence crashes from 0.49$\rightarrow$0.32$\rightarrow$0.23$\rightarrow$0.17—matches the onset of the collapse in jailbreak success.
> This provides causal evidence: collapse occurs because the model can no longer reconstruct or interpret the perturbed query.
>
> 3. When refusals occur vs. when they disappear
>
> Your suggestion also helped us isolate failure modes:
>
> Rise region:
> Recoverability, BERTScore, and confidence remain high $\rightarrow$ the model clearly understands the harmful meaning $\rightarrow$ failures are explicit refusals.
>
> Collapse region:
> Semantic metrics degrade sharply $\rightarrow$ refusals disappear completely $\rightarrow$ failures are due to misinterpretation or incoherence, not safety activation.
>
> Thus, the ASR drop at high-k comes from semantic collapse, not alignment.
>
> 4. Why GCG does not contradict the above
>
> GCG perturbs only a suffix, leaving the core harmful query intact and fully interpretable.
> In contrast, token-form drift perturbs the entire query, directly harming reconstructability.
>
> Thus:
>
> (i)GCG nonsensical ≠ drift nonsensical
>
> (ii)GCG preserves token identity of harmful content $\rightarrow$ semantic pathway intact
>
> (iii)Drift destroys token identity globally $\rightarrow$ semantic pathway collapses
>
> (iiii)GCG bypasses safety without reducing interpretability; collapse arises because interpretability is lost.
> These are fundamentally different mechanisms.

---

> ### Author Response · Authors · 2025-11-26
>
> Weakness 4: Seed design and contribution of evolution
>
> You raises two related concerns: whether the perturbation seeds are manually designed, and how much ASR comes from seeds versus later evolution. First, the seeds are not hand-crafted. As stated in line 310 (“GPT-5 was prompted to produce initial seeds”), the system starts from GPT-5–generated seeds, and the pipeline is fully automated.
>
> Regarding how much ASR comes from the initial seeds versus the later evolved forms:
> different seeds evolve into structurally different “form families”, so ASR at a specific mutation step is not directly comparable across runs. For this reason, the most meaningful and stable comparison is between (i) seed-only ASR and (ii) final ASR after the full mutation-and-selection process.
>
> Under this lens, we find a clear pattern. On large open-source models (e.g., Llama-3.1-70B) and all black-box models (GPT-5, Claude-4.), seed-only ASR is consistently low (typically <20\%), and high ASR (≥80\%) is achieved only after the evolutionary process. In contrast, for some smaller open-source models, certain GPT-5–generated seeds (e.g., double-consonant drift) already attain moderately high ASR, and evolution mainly serves to refine them beyond our 80\% threshold.
>
> Although per-mutation steps are not directly comparable across seeds, we agree that their overall trend is informative. In the revision, we will therefore (i) report seed ASR vs. final ASR explicitly, and (ii) add per-iteration ASR curves showing that evolved forms consistently dominate seed performance, and that evolution is essential for large and black-box models.
>
> Taken together, this analysis shows that our attack does not rely on manually crafted seeds, and that the core strength of the framework lies in its automated discovery and refinement of increasingly effective drift forms, rather than in the initial conditions.
>
> Question 1: Semantic preservation and consistency between attacked model and evaluator
>
> For the concern about the consistency of “semantic preservation” between the attacked model and the evaluator, our framework is designed to avoid any reliance on shared internal representations. The evaluator LLM does not directly interpret or reverse the perturbed text produced by the attacked model. Instead, all restoration is performed by a deterministic Python procedure that exactly inverts the corresponding token-form drift rule (e.g., consonant-doubling removal). This inverse mapping is rule-based and does not require any semantic inference.
>
> The evaluator therefore receives only the restored English text, not the perturbed text. Its task is simply to judge (i) whether the restored output is harmful and (ii) whether it is coherent. Because the evaluator operates exclusively on fully restored natural-language strings, its semantic space need not match that of the attacked model.
>
> This mechanism also ensures alignment between the evaluator’s judgment and the attacked model’s understanding: if the attacked model fails to follow the transformation rule, the restoration step produces garbled or inconsistent output, which is automatically marked as an attack failure. Conversely, when the attacked model correctly follows the transformation, the restoration yields fluent English directly reflecting the model’s underlying interpretation.
>
>
> Question 2: Dependence on tokenization granularity
>
> To further confirm that the rise–plateau–collapse phenomenon is not an artifact of a specific tokenizer, we evaluated the same token-form drift procedure on Gemma-3-it-27B on AdvBench and JBB, which uses a SentencePiece tokenizer.
>
> | replace | success rate on advBench% | success rate on JBB% |
> |-:|--:|--:|
> | 0| 0.0 | 0.0 |
> | 1| 83.0| 77.0|
> | 2| 86.0| 85.0|
> | 3| 90.0| 84.0|
> | 4| 96.0| 94.0 |
> | 5| 90.0| 95.0|
> | 6| 92.0| 94.0  |
> | 7  | 90.0  | 93.0  |
> | 8  | 94.0  | 91.0  |
> | 9  | 86.0  | 88.0  |
> | 10 | 80.0  | 81.0  |
> | 11 | 76.0  | 73.0  |
> | 12 | 63.0  | 61.0  |
> | 13 | 41.0  | 33.0  |
> | 14 | 23.0  | 14.0  |
> | 15 | 10.0  | 3.0   |
> | 16 | 0.0   | 0.0   |
> | 17 | 0.0   | 0.0   |
> | 18 | 0.0   | 0.0   |
> | 19 | 0.0   | 0.0   |
> | 20 | 0.0   | 0.0   |
> | 21 | 0.0   | 0.0   |
> | 22 | 0.0   | 0.0   |
> | 23 | 0.0   | 0.0   |
> | 24 | 0.0   | 0.0   |
> | 25 | 0.0   | 0.0   |
> | 26 | 0.0   | 0.0   |
>
>
> The resulting curve is consistent with other models:
>
> success rate on advBench rises sharply from 0\% $\rightarrow$ 96\% (k = 1–4),
>
> remains in a high plateau ($\approx$80–95\% for k = 5–11), and
>
> collapses to 0\% once k $\ge$ 16.
>
> Despite its distinct tokenization scheme and alignment pipeline, Gemma-3 exhibits the same transition dynamics. This strongly indicates that the phenomenon arises from semantic degradation under token-form drift, rather than from tokenizer boundaries or any model-specific artifact. The replication across open-source and black-box models, across architectures, and now across tokenizers provides compelling evidence that this behavior is systematic and architecture-agnostic.

---

> ### Author Response · Authors · 2025-11-26
>
> Question 3: Stronger alignment (beyond SFT) and robustness under PPO/GRPO
>
> Regarding the question on more strongly aligned models, we conducted additional experiments where we fine-tuned Qwen-2.5-72B using PPO and GRPO with our safety datasets and hyperparameters similar to those used in prior RLHF work. Despite these strengthened alignment stages, token-form drift remained highly effective:
>
> PPO-aligned model:
> initial drift “double consonant” achieves 0\% ASR;
> after one evolution step (“double consonant + number replacement”), ASR increases to 80\%.
>
> GRPO-aligned model:
> initial drift achieves 10\% ASR;
> after one evolution step (“double consonant + vowel shifting”), ASR rises to 90\%.
>
> This shows that RL-based alignment, which is known to correct superficial refusal behaviors, does not eliminate drift vulnerabilities; the attack quickly adapts by discovering alternative surface-form shifts.
>
> Furthermore, our black-box results provide additional evidence: both GPT-5 and Claude, which undergo multi-stage proprietary alignment pipelines (including SFT + RLHF-style optimization), are still vulnerable to token-form drift, despite being substantially more aligned than open-source models. Their initial seed ASR is below 20\%, yet evolved forms consistently surpass our 80\% threshold.

---

### Meta-Review · Area_Chair_m7pk · 2026-01-08

**Summary:**

While the paper identifies an interesting phenomenon regarding "token-form drift," reviewers raised consistent concerns regarding the intellectual novelty and technical depth of the work that inform a rejection decision. The paper's central observation (e.g., safety alignment is sensitive to token shifts) is a finding many practitioners are already familiar with, and the "rise-plateau-collapse" pattern mirrors results seen in prior cipher-based and encoding attack studies. Furthermore, the manuscript does not deliver a sufficiently concrete technical contribution, as the proposed automated framework is conceptually similar to existing mutation-based prompt optimization methods like AutoDAN. Reviewers were concerned that the analysis provided limited new mechanistic or theoretical insight to distinguish it from a systematic evaluation of known vulnerabilities.

**Reviewer Concerns:**

The rebuttal helped clarify that the automated framework is intended as a diagnostic tool and provided empirical evidence that the observed collapse is due to semantic degradation rather than intent detection. However, it did not meaningfully resolve core issues regarding limited technical novelty and the lack of a principled model for the phenomenon. Additionally, the paper fails to propose or validate a concrete defense, leaving the work as a purely observational study without a clear path toward mitigation. Consequently, reviewers' fundamental doubts regarding the paper's contribution to the venue still remain.

**Reviewer Scores:**

No score increase is expected, as the concerns regarding limited novelty and technical contribution remain.

---

### Decision · Program_Chairs · 2026-01-26

Reject